# Structure and mechanistic features of the prokaryotic minimal RNase P

**Rebecca Feyh[1†], Nadine B Waeber[1†], Simone Prinz[2], Pietro Ivan Giammarinaro[3], Gert Bange[3,4], Georg Hochberg[3,4], Roland K Hartmann[1]\*, Florian Altegoer[3]\***

[1]Institute of Pharmaceutical Chemistry, Philipps-University Marburg, Marburg, Germany; [2]Department of Structural Biology, Max Planck Institute of Biophysics, Frankfurt, Germany; [3]Center for Synthetic Microbiology and Department of Chemistry, Philipps-University Marburg, Marburg, Germany; [4]Max-Planck Institute for Terrestrial Microbiology, Marburg, Germany

**Abstract** Endonucleolytic removal of 5'-leader sequences from tRNA precursor transcripts (pre-tRNAs) by ribonuclease P (RNase P) is essential for protein synthesis. Beyond RNA-based RNase P enzymes, protein-only versions of the enzyme exert this function in various eukarya (there termed PRORPs) and in some bacteria (*Aquifex aeolicus* and close relatives); both enzyme types belong to distinct subgroups of the PIN domain metallonuclease superfamily. Homologs of *Aquifex* RNase P (HARPs) are also expressed in some other bacteria and many archaea, where they coexist with RNA-based RNase P and do not represent the main RNase P activity. Here, we solved the structure of the bacterial HARP from *Halorhodospira halophila* by cryo-electron microscopy, revealing a novel screw-like dodecameric assembly. Biochemical experiments demonstrate that oligomerization is required for RNase P activity of HARPs. We propose that the tRNA substrate binds to an extended spike-helix (SH) domain that protrudes from the screw-like assembly to position the 5'-end in close proximity to the active site of the neighboring dimer. The structure suggests that eukaryotic PRORPs and prokaryotic HARPs recognize the same structural elements of pre-tRNAs (tRNA elbow region and cleavage site). Our analysis thus delivers the structural and mechanistic basis for pre-tRNA processing by the prokaryotic HARP system.

**\*For correspondence:**
roland.hartmann@staff.uni-marburg.de (RKH);
altegoer@uni-marburg.de (FA)

[†]These authors contributed equally to this work

**Competing interests:** The authors declare that no competing interests exist.

## Introduction

Ribonuclease P (RNase P) is the essential endonuclease that catalyzes the 5'-end maturation of tRNAs (*Klemm et al., 2016*; *Rossmanith and Hartmann, 2020*; *Guerrier-Takada et al., 1983*). The enzyme is present in all forms of life, yet shows a remarkable variation in the molecular architecture. There are two basic types of RNase P, RNA-based and protein-only variants. The former consists of a structurally conserved, catalytic RNA molecule that associates with a varying number of protein cofactors (1 in bacteria, 5 in archaea, and 9–10 in eukarya [*Klemm et al., 2016*; *Jarrous and Gopalan, 2010*]). Protein-only enzymes arose independently twice in evolution. In eukarya, a protein-only RNase P (termed PRORP) apparently originated at the root of eukaryotic evolution and is present in four of the five eukaryotic supergroups (*Lechner et al., 2015*). This type of enzyme replaced the RNA-based enzyme in one compartment or even in all compartments with protein synthesis machineries, such as land plants harboring PRORP enzymes in the nucleus, mitochondria, and chloroplasts (*Gobert et al., 2010*). In metazoan mitochondria, PRORP requires two additional protein cofactors for efficient function (*Holzmann et al., 2008*).

More recently, a bacterial protein-only RNase P, associated with a single polypeptide as small as ~23 kDa, was discovered in the hyperthermophilic bacterium *Aquifex aeolicus* that lost the genes for the RNA and protein subunits (*rnpB* and *rnpA*) of the classical and ancient bacterial RNase P (*Nickel et al., 2017*). This prokaryotic type of minimal RNase P system was named HARP (for:

Homolog of Aquifex RNase P) and identified in 5 other of the 36 bacterial phyla beyond Aquificae (*Nickel et al., 2017*). Among HARP-encoding bacteria, only some lack the genes for RNA-based RNase P (i.e., Aquificae, Nitrospirae), while others harbor *rnpA* and *rnpB* genes as well (*Daniels et al., 2019*; *Nickel et al., 2017*). Overall, HARP genes are more abundant in archaea than bacteria. However, all of these HARP-positive archaea also encode the RNA and protein subunits of the RNA-based RNase P (*Nickel et al., 2017*; *Daniels et al., 2019*). Remarkably, HARP gene knock-outs in two Euryarchaeota, *Haloferax volcanii* and *Methanosarcina mazei*, showed no growth pheno-types under standard conditions, temperature, and salt stress (*H. volcanii*) or nitrogen deficiency (*M. mazei*) (*Schwarz et al., 2019*). In contrast, it was impossible to entirely erase the RNase P RNA gene from the polyploid genome of *H. volcanii* (~18 genome copies per cell in exponential growth phase; *Breuert et al., 2006*). Even a knockdown to ~20% of the wild-type RNase P RNA level in *H. volcanii* was detrimental to tRNA processing and resulted in retarded cell growth (*Stachler and March-felder, 2016*). The findings suggest that HARP is neither essential nor represents the housekeeping RNase P function in archaea, explaining its sporadic loss in archaea. HARPs are evolutionarily linked to toxin-antitoxin systems (*Daniels et al., 2019*; *Schwarz et al., 2019*; *Gobert et al., 2019*). Fre-quently, the toxin proteins are endoribonucleases that cleave mRNA, rRNA, tmRNA, or tRNA to inhibit protein biosynthesis in response to certain stresses (*Masuda and Inouye, 2017*). Conceivably, the progenitor of *A. aeolicus* and related Aquificaceae might have acquired such a toxin-like tRNA endonuclease via horizontal gene transfer and established it as the main RNase P activity with rela-tively little reprogramming.

HARPs belong to the PIN domain-like superfamily of metallonucleases. They were assigned to the PIN_5 cluster, VapC structural group, whereas eukaryal PRORPs belong to a different subgroup of this superfamily (*Matelska et al., 2017*; *Gobert et al., 2019*). HARPs oligomerize and Aq880 was originally observed to elute as a large homo-oligomeric complex of ~420 kDa in gel filtration experi-ments (*Nickel et al., 2017*). However, its specific mode of substrate recognition and the underlying structural basis is lacking to date. Here, we present the homo-dodecameric structure of the HARP from the γ-bacterium *Halorhodospira halophila* SL1 (Hhal2243) solved by cryo-electron microscopy (EM) at 3.37 Å resolution. Furthermore, we employed mass photometry (MP) to investi-gate the oligomerization behavior of HARP and correlated the oligomeric state with enzyme activity. Our structure reveals that HARPs form stable dimers via a two-helix domain inserted into the metal-lonuclease domain. These dimers further assemble into a screw-like assembly resulting in an asym-metric and thus imperfect novel type of homo-dodecamer. Our biochemical analysis suggests that pre-tRNA processing involves the neighboring dimers and thus requires the formation of a higher-order HARP oligomer. In conclusion, we here present the structural basis for the RNase P-like pre-tRNA processing activity of prokaryotic HARPs.

## Results

### Dodecameric structure of the HARP from *H. halophila*

Structural information on *A. aeolicus* RNase P (Aq880) and HARPs is lacking and a mechanistic understanding of pre-tRNA processing by HARPs has remained unknown so far. As previously observed by size exclusion chromatography (SEC), Aq880 forms large oligomers of ~420 kDa (*Nickel et al., 2017*). Our attempts to resolve its structure by X-ray crystallography or nuclear mag-netic resonance were unsuccessful. Therefore, we also purified other HARPs to increase the chances of successful structure determination. Among those was the HARP of *H. halophila* (Hhal2243) that was purified to homogeneity using a two-step protocol consisting of $Ni^{2+}$-affinity chromatography and anion-exchange chromatography (*Figure 1—figure supplement 1*; see Materials and methods). Hhal2243 formed an assembly of similar size as Aq880 (*Figure 1—figure supplement 2A*) but adopted a more uniform oligomeric state than Aq880 (see below). Like Aq880 (*Marszalkowski et al., 2008*), Hhal2243 showed pre-tRNA processing activity in the presence of $Mg^{2+}$ and $Mn^{2+}$ (*Figure 1—figure supplement 2B,C*).

We succeeded in solving the structure of Hhal2243 by cryo-EM to 3.37 Å resolution (*Figure 1*, *Figure 1—figure supplement 3*, *supplementary file 1a*). Hhal2243 assembles into a dodecamer in a left-handed screw-like manner, with each molecule being rotated by approximately 58°, except for the interface between dimers 1/1* and 6/6*. Upon completion of the single screw turn, the first and

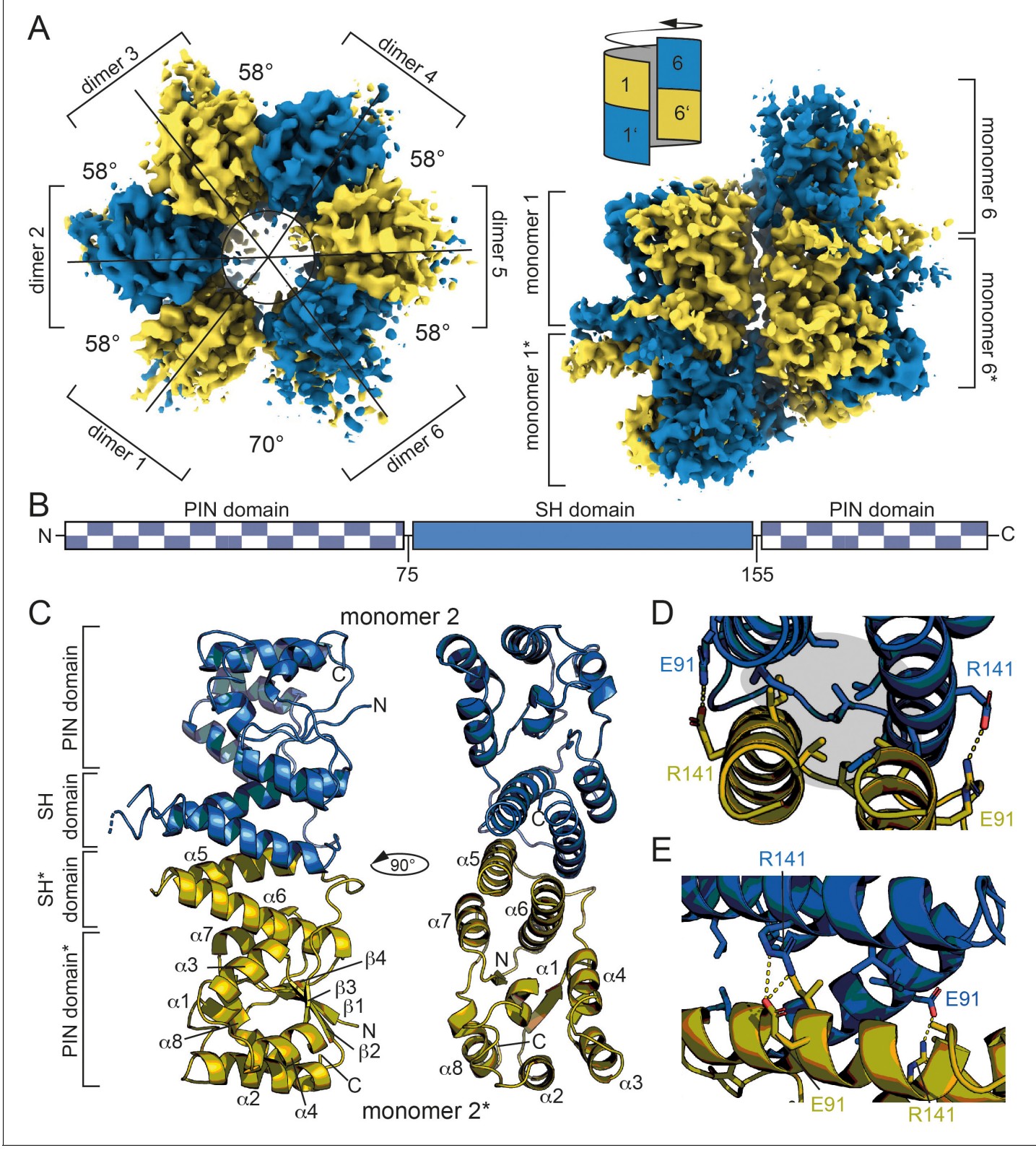

**Figure 1.** Dodecameric structure of Hhal2243. (**A**) Cryo-EM electron density map of Hhal2243 shown from a top view (left) and a side view (right). The monomers are colored in blue and olive, respectively. The angles between dimers are indicated in the top view. The sketch in the upper left corner of the view on the right indicates how the dimers assemble to form the screw-like arrangement of the dodecamer. (**B**) Domain architecture of Hhal2243. (**C**) Model of the Hhal2243 dimer. The protein consists of a PIN five domain with an inserted spike-helix (SH) domain forming the dimer interface. The

*Figure 1 continued on next page*

*Figure 1 continued*

numbers indicate the amino acid boundaries of the SH domain. (**D, E**) Detailed view of the dimer interface. The clamping salt bridges are shown as sticks and indicated with dashed yellow lines. The hydrophobic core between the two monomers is marked by the gray sphere in (**D**). EM, electron microscopy.

The online version of this article includes the following source data and figure supplement(s) for figure 1:

**Figure supplement 1.** Purification of Hhal2243.

**Figure supplement 1—source data 1.** SDS-PAGEs and Western blot from the purification of Hhal2243.

**Figure supplement 2.** Analysis of Hhal2243.

**Figure supplement 2—source data 1.** RNAse P processing of 5′-$^{32}$P-labeled T. thermophilus pre-tRNA$^{Gly}$ by Aq880 and Hhal2243.

**Figure supplement 3.** Cryo-EM data collection and analysis.

**Figure supplement 4.** Hypothetical geometric scenarios of HARP oligomerization.

last dimers (1/1* and 6/6*) encounter each other at a slightly larger angle of 70° (*Figure 1A*, top view, left) and laterally displaced (*Figure 1A*, side view, right); they cannot form the same inter-dimer contacts as dimers can form within the screw turn; the steric clash between dimers 1/1* and 6/6* terminates oligomerization. Hypothetic geometric scenarios that would allow a continuation of oligomerization beyond a dodecamer are illustrated in *Figure 1—figure supplement 4*. Hhal2243 consists of a PIN-like metallonuclease domain into which two helices are inserted that we termed the 'spike-helix' (SH) domain (*Figure 1B*). The PIN-like domain is formed by six α-helices (α1–α4, α7, and α8) and four β-strands (β1–β4) that fold into an α/β/α domain with a central, four-stranded parallel β-sheet (*Figure 1C*). Two Hhal2243 monomers align head-to-head with their SH domains consisting of helices α5 and α6 to form a dimer, while two SH domains form a four-helix bundle resulting in six spikes that protrude from the dodecameric assembly (*Figure 1C*). The dimer interface covers a buried surface area of 1300 Å$^2$ and is mainly of hydrophobic nature with two clamping salt bridges formed by R141 and E91 from either monomer, respectively (*Figure 1D,E*).

## Oligomerization and its influence on pre-tRNA processing activity

Our findings and the conservation of HARPs suggested that the dodecameric superstructure represents a conserved feature of HARPs and might therefore be required for their RNase P-like activity. To scrutinize this idea, we took a closer look at the interactions between two dimers. The covered interface between neighboring dimers extends over 900 Å$^2$ and involves the long α4–α5 loop of one monomer and the α7–α8 loop as well as helix α8 of the respective other monomer (*Figure 2A*). The interactions between interdimer residues are mainly of polar nature.

To investigate the oligomerization behavior, we employed MP, a method allowing for the rapid and reliable determination of the dynamic oligomeric distribution of macromolecules in solution (*Sonn-Segev et al., 2020*; *Soltermann et al., 2020*). Application of this method to Hhal2243 revealed a stable dodecameric assembly of 295 kDa that included 98% of all molecules in the sample (*Figure 2B*, top left panel). To compare this oligomerization behavior with that of Aq880, we purified Aq880 by Ni-ion affinity chromatography and SEC (*Figure 2—figure supplements 1* and *2*). Interestingly, despite their similar behavior on SEC, MP of Aq880 unveiled a major species at 277 kDa that, however, included only 47% of all molecules, while several subspecies with lower molecular weights became visible (*Figure 2B*, top right panel; *supplementary file 1b*). This polydispersity of Aq880, not detectable in SEC profiles, is probably the reason why all structural approaches failed so far in the case of Aq880.

To investigate whether oligomerization of Aq880 might impact its pre-tRNA processing activity, we analyzed the oligomer interface of the Hhal2243 structure in more detail. As outlined above, the interdimer interaction involves the α4–α5 loop of one monomer (e.g., monomer 1) and the α7–α8 loop as well as the C-terminal helix α8 of the other monomer (e.g., monomer 2) provided by the neighboring dimer (*Figure 2A*). This implied that C-terminal protein truncations affecting the integrity of helix α8 will impede oligomer formation. We thus evaluated the ability of C-terminally truncated Aq880 variants to oligomerize by MP and their activity in pre-tRNA processing experiments. Five truncated variants of Aq880 were purified to homogeneity and analyzed by SEC (*Figure 2—figure supplement 2*). Notably, SEC calibration suggested molecular masses of 55–68 kDa for all truncated Aq880 variants (*Figure 2—figure supplement 2*). MP of protein variant Aq880$_{\Delta184–191}$ still

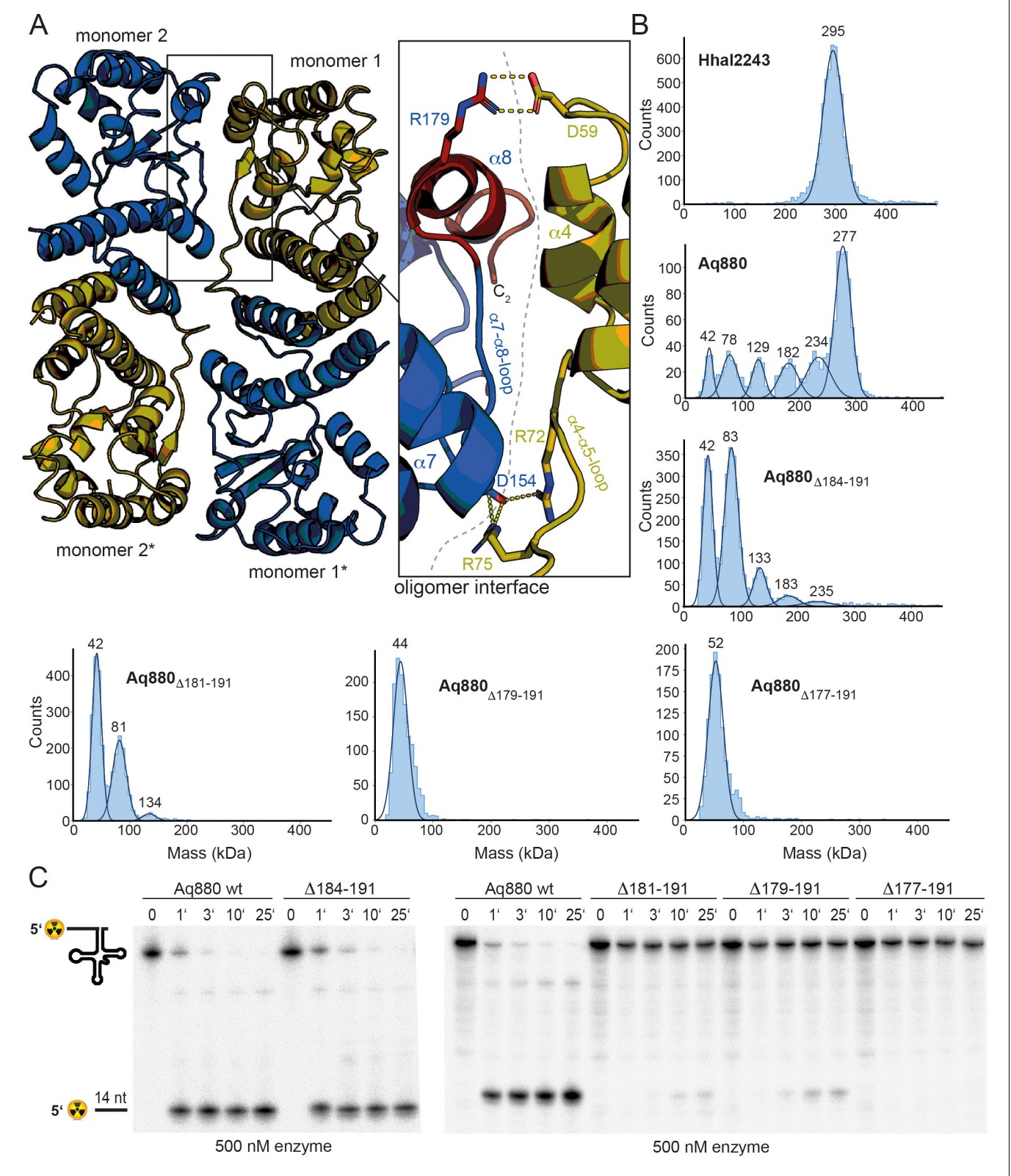

**Figure 2.** Oligomerization is required for HARP activity. (**A**) Oligomer interface between two monomers (colored in blue and olive) of interacting dimers. The C-terminus starting at residue 177 is colored in red to highlight the region deleted in the 'shortest' Aq880 variant (Δ177–191). Salt bridges are indicated with yellow dashed lines. (**B**) Mass photometry of Hhal2243, Aq880 wt, Aq880$_{\Delta184–191}$, Aq880$_{\Delta181–191}$, Aq880$_{\Delta179–191}$, and Aq880$_{\Delta177–191}$. Molecular masses corresponding to the respective Gaussian fits are shown in kDa above the fits. (**C**) Processing of pre-tRNA$^{Gly}$ by Aq880 wt and

*Figure 2 continued on next page*

*Figure 2 continued*

derived mutant variants. Aliquots were withdrawn at different time points (1, 3, 10, or 25 min) of incubation at 37°C; 0, substrate without addition of enzyme. Aq880 wt in comparison with the C-terminal deletion variants Δ177–191, Δ179–191, Δ181–191, and Δ184–191, all at 500 nM enzyme. For more details, see Materials and methods. Source data of phosphor images are available in *Figure 2—source data 1*.

The online version of this article includes the following source data and figure supplement(s) for figure 2:

**Source data 1.** Processing of pre-tRNA^Gly by Aq880 wt and derived truncated variants.

**Figure supplement 1.** Purification of Aq880.

**Figure supplement 1—source data 1.** SDS-PAGEs from the purification of Aq880.

**Figure supplement 2.** Chromatographic analysis of Aq880 WT and C-terminal deletion mutants.

showed a considerable subfraction (~18%; see *supplementary file 1b*) of higher-order oligomers ($\geq$133 kDa), whereas Aq880$_{\Delta181-191}$ formed only 4% hexamer (peak at 134 kDa), 41% tetramer (peak at 81 kDa) beyond the dimer subfraction at 42 kDa (*Figure 2B*, middle panels; *supplementary file 1b*). Further truncation of Aq880 had an impact on protein stability, as judged by lower purification yields, and MP revealed that variants Aq880$_{\Delta179-191}$ and Aq880$_{\Delta177-191}$ form only dimeric species (*Figure 2B*, lower panels). Processing assays revealed that deletion of residues 184–191 resulted in a protein that still had substantial activity, while all the other truncations showed almost no activity or were entirely inactive (*Table 1*, *Figure 2C*). Thus, our experiments show that the activity of the enzyme in pre-tRNA processing assays depends on its ability to form higher-order oligomers.

## The active site is conserved among HARPs and PRORPs

Phylogenetic analyses indicate that the two protein-only RNase P systems found in bacteria and eukarya evolved independently (*Lechner et al., 2015*; *Nickel et al., 2017*). This is consistent with the different structural basis of the two types of enzymes acting on tRNAs. PRORPs are composed of a PPR domain important for substrate recognition, a central zinc finger domain, and a flexible hinge connecting it to the metallonuclease domain (*Howard et al., 2012*; *Teramoto et al., 2020*). In contrast, HARPs solely consist of a metallonuclease domain (*Figure 3A*). The active sites of both molecules superpose well with a root mean square deviation (r.m.s.d.) of 0.445 over 30 Cα atoms. More precisely, the β-strands 5, 6, and 8, and the α-helices 16 and 20 within the metallonuclease domain of *Arabidopsis thaliana* PRORP1 (*At*PRORP1) were used for the structural superposition with β1, β4, α1, and α6 of Hhal2243, as the overall scaffold of the two metallonuclease domains is related but shows large structural deviations. In earlier studies, we could already show that the catalytic aspartates D7, D138, D142, and D161 are indispensable for Aq880 activity (*Nickel et al., 2017*; *Schwarz et al., 2019*). Our Hhal2243 structure supports the prediction that these residues constitute the active site of the protein (*Figure 3B*), including an almost perfect superposition with three of the

**Table 1.** Processing activities of Aq880 variants.
n.d., no cleavage detectable.

| Aq880 variant | kobs (min-1) ($\pm$SD) | Substrate cleaved after 25 min (in %) |
|---|---|---|
| wt 50 nM wt 500 nM | 0.62 ± 0.15<br>2.06 ± 0.42 | 95<br>98 |
| Δ184–191 50 nM<br>Δ184–191 500 nM | 0.22 ± 0.08<br>1.41 ± 0.07 | 46<br>97 |
| Δ181–191 500 nM<br>Δ179–191 500 nM | $(4 \pm 1) \times 10^{-3}$<br>$(7 \pm 3) \times 10^{-3}$ | 9<br>15 |
| Δ177–191 500 nM | – | n.d. |
| R125A 50 nM<br>R125A 500 nM | $(15 \pm 3) \times 10^{-4}$<br>$(16.1 \pm 0.3) \times 10^{-2}$ | 2<br>26 |
| R129A 500 nM | – | n.d. |
| R125A/R129A 500 nM | – | n.d. |
| K119A/R123A/R125A/K127A/R129A 500 nM | – | n.d. |

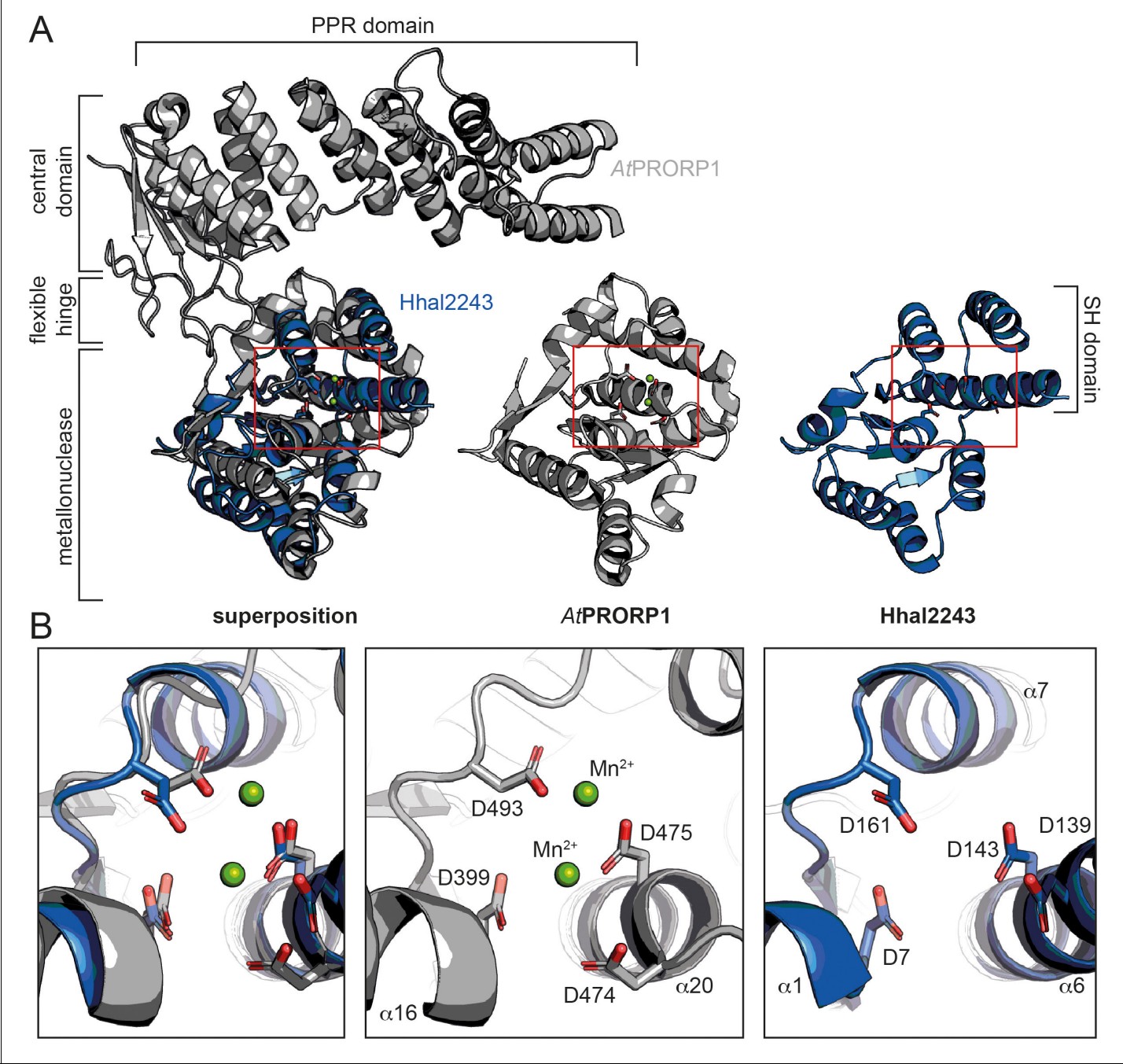

**Figure 3.** Comparison between HARP and PRORP systems. (**A**) Superposition of AtPRORP1 (PDB: 4G24) and Hhal2243 shows that the metallonuclease domain superposes well, while the PPR domain is absent in HARPs. The red rectangle shows the closeup in (B). (**B**) Orientation of active site residues is conserved among PRORPs and HARPs; the active site residues are positioned similarly in both enzyme types. AtPRORP1 is colored in gray and Hhal2243 in blue.

four active site aspartates of *At*PRORP1 (D399, D475, and D493; *Howard et al., 2012*; *Figure 3B*). We conclude that the mechanism of catalysis is conserved among the two distinct protein families with RNase P activity.

## The SH domain is critical for RNase P activity

In the next step, we sought to identify those amino acid residues that are required for the interaction of the enzyme with its pre-tRNA substrate. Helices α5 and α6 of the SH domain expose several conserved arginines and lysines that might be critical for pre-tRNA binding. Notably, several of the residues lie within the distal part of the SH domain that was not resolved in our cryo-EM structure, likely due to a high degree of conformational flexibility. As secondary structure predictions indicated a continuation of the helical arrangement in this region, we generated a homology model of Aq880 based on the Hhal2243 structure and the secondary structure prediction (*Figure 4—figure supplement 1A*). The resulting Aq880 model was further verified by rigid-body docking into the electron density map of our Hhal2243 structure (*Figure 4—figure supplement 1B*).

All residues that were varied to alanines are within the distal part of helix α6 of the SH domain (*Figure 4A*). Variation of the entire positive arginine/lysine stretch resulted in the inability of the mutant protein to cleave off the 5'-leader sequence (*Figure 4B*). To narrow down the critical residues, we generated the R125A/R129A double mutant and protein variants with single mutations of R125 or R129. While both R129A and the double mutant R125A/R129A were inactive, R125A retained residual activity (*Figure 4C*, *Table 1*). Notably, variation of positively charged residues to alanines in the SH domain did not significantly change the oligomerization behavior of Aq880 as judged by MP (*Figure 4D*). Our data suggest that the cluster of positively charged side chains in the SH domain is required for pre-tRNA binding. Although this part is less ordered in our cryo-EM structure, it might become more rigid upon tRNA substrate binding.

## Discussion

Here, we present the cryo-EM structure of Hhal2243, a member of the recently described HARP family of bacterial and archaeal proteins with RNase P activity (*Nickel et al., 2017*). Our combined structural and biochemical analysis sheds light on this prokaryotic minimal protein-only RNase P system.

The Hhal2243 HARP structure assembles into a homo-dodecameric ultrastructure. The dodecamer consists of six dimers that oligomerize in a screw-like fashion (*Figure 1*). The monomeric subunits of the dimers interact via their SH domains. There are many examples of proteins forming symmetric homo-dodecameric assemblies (e.g., glutamine synthase [*van Rooyen et al., 2011*]; helicases [*Bazin et al., 2015*] or bacterial DNA-binding proteins expressed in the stationary phase (DPS) [*Roy et al., 2008*]) and recent studies have discussed the theoretical types of possible quaternary structures (*Laniado and Yeates, 2020*; *Ahnert et al., 2015*). However, the screw-like assembly of HARPs leads to an asymmetric and thus imperfect novel type of oligomer. This way, HARPs form a defined quaternary structure that is not entirely symmetric, while the screw-like assembly terminates the incorporation of new subunits at the stage of a dodecamer. By stepwise truncation of the C-terminus of Aq880, oligomerization could be reduced or abolished, which correlated with functional losses (*Figure 2C*). Our results thus clearly show that only oligomeric species of HARP are able to cleave the 5'-leader sequence of pre-tRNAs.

HARPs belong to the PIN domain-like superfamily of metallonucleases and share this classification with the eukaryotic PRORPs, although the two systems belong to distinct subgroups (*Matelska et al., 2017*). We compared our structure to the PIN domain of PRORPs and could demonstrate that the active site residues superpose well (*Figure 3B*), suggesting that the catalytic mechanism is basically conserved. Notably, the two proteins had to be superposed over a small range of Cα−atoms to obtain a good r.m.s.d. (*Figure 3A*) owing to the large overall differences of the two PIN domains.

With the knowledge of the structural conservation of the active site, we examined the pre-tRNA substrate recognition and binding by the two protein-only RNase P systems: the crystal structure of the PPR domain of *At*PRORP1 in complex with a tRNA unveiled how PRORPs recognize the pre-tRNA via its PPR RNA-binding domain (*Teramoto et al., 2020*). The characteristic feature of classical pentatricopeptide repeats (PPRs) is that each PPR repeat recognizes a specific nucleotide, allowing high RNA target specificity (*Yan et al., 2019*). Interestingly, the PPR domain of PRORP recognizes conserved structural elements of tRNAs rather than single nucleotides in a binding mode reminiscent to that of the RNA-based RNase P holoenzyme (*Teramoto et al., 2020*). More precisely, the elbow region, where D- and T-loop interact, is the main tRNA-protein interaction interface (*Teramoto et al., 2020*), a feature that became already evident in previous biochemical and

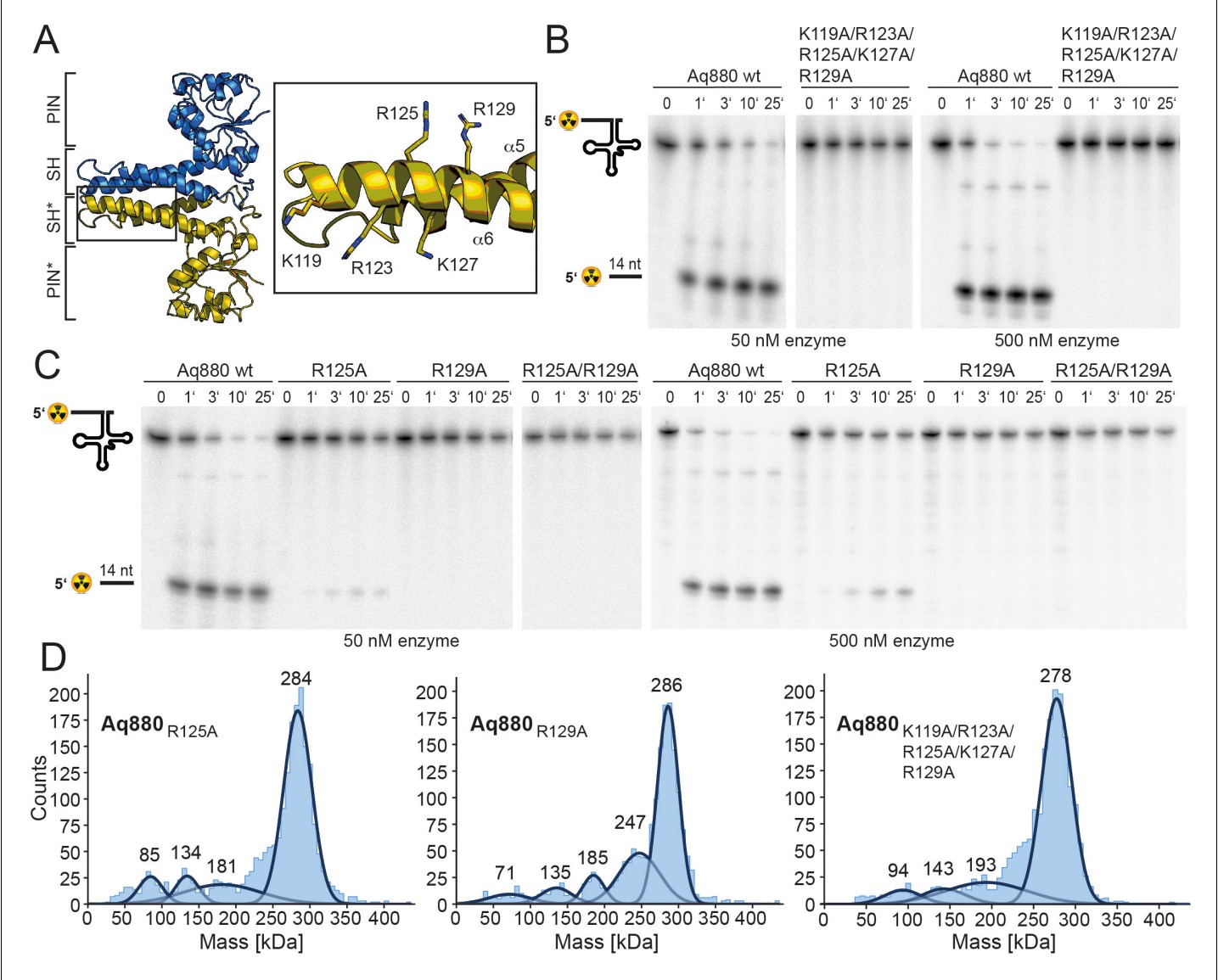

**Figure 4.** The SH domain is essential for RNase P activity. (**A**) Homology model of Aq880 with residues critical for pre-tRNA processing activity in the SH domain displayed as sticks. Processing of pre-tRNA$^{Gly}$ by Aq880 wt and derived mutant variants. Aliquots were withdrawn at different time points (1, 3, 10, or 25 min) of incubation at 37°C; 0, substrate without addition of enzyme. (**B**) Aq880 wt and the quintuple mutant K119A/R123A/R125A/K127A/R129A at 50 nM (left) or 500 nM (right) enzyme; (**C**) Aq880 wt, the single mutants R125A and R129A, and the double mutant R125A/R129A, assayed at 50 nM (left) or 500 nM (right) enzyme. Source data of phosphor images are available in *Figure 4—source data 1*. (**D**) Mass photometry of Aq880$_{R125A}$, Aq880$_{R129A}$, and Aq880$_{K119A/R123A/R125A/K127A/R129A}$. Molecular masses corresponding to the respective Gaussian fits are shown in kDa above the fits. Source data of phosphor images are available in *Figure 4—source data 1*. SH, spike helix.

The online version of this article includes the following source data and figure supplement(s) for figure 4:

**Source data 1.** Processing of pre tRNA$^{Gly}$ by Aq880 wt and derived truncated mutant variants.

**Figure supplement 1.** Homology model generation of Aq880.

biophysical investigations (*Pinker et al., 2013*; *Gobert et al., 2013*; *Brillante et al., 2016*; *Pinker et al., 2017*). Moreover, the same region, representing the most conserved structural element of tRNAs, is also recognized by RNA-based RNase P enzymes (reviewed in *Rossmanith and Hartmann, 2020*). We thus considered it likely that pre-tRNA recognition by HARPs involves the tRNA elbow region in a similar manner as for all other RNase P types, although we were puzzled by the absence of any evident RNA binding motifs in HARPs. We then focused on exposed positively charged residues and were indeed able to identify a stretch of conserved arginine and lysine

residues within the SH domain of the protein. Variation of this positive stretch to alanines rendered the protein inactive (*Figure 4*).

Combining our knowledge on the active site, on the residues critical for pre-tRNA binding and the correlation of an oligomeric assembly with enzymatic activity, we set out to propose a model for pre-tRNA binding by HARP proteins. Assuming that the outer SH domain, harboring the stretch of positively charged amino acid side chains, interacts with the tRNA elbow, then the distance of approximately 45 Å between the outer SH domain of one dimer and the active site of the neighboring dimer perfectly positions one pre-tRNA molecule for cleavage (*Figure 5A*). To test this, we used the yeast tRNA from the PRORP structure as model (*Teramoto et al., 2020*) and could perfectly dock it onto our HARP structure (*Figure 5B*). Notably, in this scenario, the D-loop is near helix α3 (*Figure 5B*). We thus also consider it likely that residues within α3 are involved in the coordination of the D-loop. Our proposed tRNA binding mode considers that oligomerization is strictly required for HARP activity. This framework enables the cooperation of two adjacent dimers, where one monomer of each dimer binds the tRNA elbow in such a manner that the pre-tRNA cleavage site, at a distance of ~45 Å, directly reaches into the active site of the monomer from the neighboring dimer (*Figure 5B*).

According to this scenario and as exemplarily illustrated in *Figure 5B*, the SH domain of monomer 6* would bind the tRNA elbow region, while the tRNA 5'-end is docked into the active site of monomer 5*. This way, five pre-tRNAs can potentially be docked onto the upper or lower layer of the dodecameric, distorted double donut structure (*Figure 5C*). This raises the question, how likely the simultaneous occupancy of all 10 potential tRNA binding sites is. So far, we have been unable to obtain stable HARP:tRNA complexes at submicromolar concentrations, for example, on size exclusion columns. However, considering that tRNA concentration within an *Escherichia coli* cell was estimated to be 0.5 mM (*Goodsell, 1991*), conditions might be conceivable (e.g., after shutdown of protein synthesis) at which all 10 tRNA binding sites could be saturated. Another possibility is cooperative tRNA binding, such that binding of one tRNA allosterically facilitates binding of the next one. Such a model could also explain the observed requirement of a higher-order oligomer structure for RNase P activity. However, we found no evidence for cooperativity in pre-tRNA cleavage kinetics catalyzed by Aq880 under multiple turnover conditions (*Nickel et al., 2017*).

Although the model is hypothetical at present, other pre-tRNA binding modes seem unlikely based on the spatial constraints imposed by the dodecameric HARP structure and the uniformity and rigidity of prokaryotic tRNAs. Our data furthermore suggest that in the higher oligomeric assemblies, the dimers mutually stabilize by each other for productive pre-tRNA processing. Variant Aq880$_{\Delta181-191}$ still forms tetramers and a low number of hexamers but is only 10% active (*Figure 2B, C*). In contrast, Aq880$_{\Delta184-191}$ also assembles into octamers, decamers, and a small subset of even dodecamers but retains over 90% activity (*Figure 2B,C*, *Table 1*). We thus consider it likely that higher oligomers form a more rigid scaffold, while a tetramer alone is too flexible for efficient binding and cleavage of precursor tRNAs. HARPs thus represent an impressive example of how a more complex biological task can be accomplished by a small protein that assembles into a large homo-oligomeric ultrastructure, and where different monomers contribute distinct partial functions.

Taken together, we here present the molecular framework for pre-tRNA processing by HARPs and show that this enzyme system, although it evolved independently of PRORPs, shares conserved features with eukaryotic PRORPs concerning pre-tRNA recognition and nuclease activity.

## Materials and methods

**Key resources table**

| Reagent type (species) or resource | Designation | Source or reference | Identifiers | Additional information |
| --- | --- | --- | --- | --- |
| Gene (*Aquifex aeolicus*) | Aq_880 | GenBank | AAC07003.1 | |
| Gene (*Halorhodospira halophila*) | Hhal_2243 | GenBank | ABM63007.1 | |

*Continued on next page*

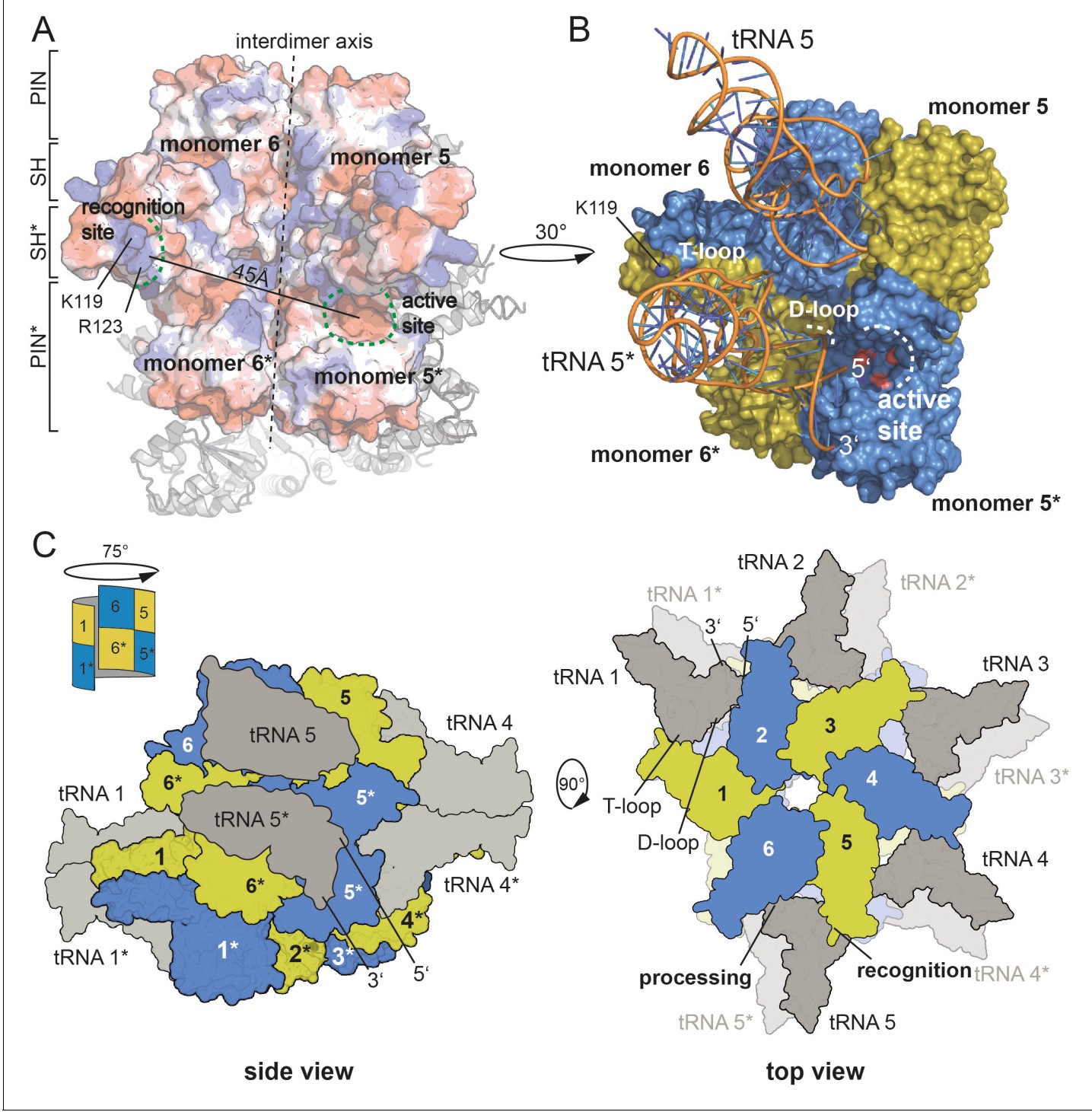

**Figure 5.** Model for tRNA recognition and processing by HARPs. (**A**) Surface of two adjacent Aq880 dimers colored according to the calculated electrostatic potential. The proposed recognition site for the tRNA elbow and the active site is indicated by curved dashed green lines; the distance between the two regions is approximately 45 Å. The almost vertical straight and dashed line marks the interdimer axis. The distance between the two regions is approximately 45 Å. The remaining subunits of the dodecamer are shown as cartoon in the background. (**B**) Closeup of the Aq880 surface colored in blue and olive. The tRNA[Phe], taken from the PRORP PPR domain co-structure (PDB: 6LVR), was docked onto our structure. The model predicts how the tRNA is coordinated via positive residues within the SH domain (of monomer 6* in this example) to position the 5'-end in close proximity to the active site of the neighboring dimer (of monomer 5* in this example). (**C**) Left: sketch of the Hhal2243 homo-dodecamer with docked tRNAs as side view. The view is rotated by 75° compared to the view in *Figure 1A* indicated by a sketch in the upper left corner. Right: sketch of tRNA recognition and cleavage by HARPs shown from the top. Numbers indicate monomers.

*Continued*

| Reagent type (species) or resource | Designation | Source or reference | Identifiers | Additional information |
|---|---|---|---|---|
| Strain, strain background (*Escherichia coli*) | Rosetta (DE3) | Merck Novagen | 70954–3 | |
| Strain, strain background (*Escherichia coli*) | Lemo21 (DE3) | NEB Biolabs | HC2528J | |
| Antibody | 6×His Tag HRP (mouse monoclonal) | Thermo Fisher Scientific, Invitrogen | Catalog #MA1-21315-HRP | Dilution 1:5000 |
| Commercial assay or kit | Gel Filtration Markers Kit for Protein Molecular Weights 12–200 kDa | Merck Sigma-Aldrich | MWGF200 | |
| Chemical compound, drug | Apoferritin | Merck Sigma-Aldrich | A3660 | |
| Software algorithm | Adobe Illustrator CC | Adobe | | Version 25.0 |
| Software algorithm | Adobe Photoshop CC | Adobe | | Version 22.0.0 |
| Software algorithm | CryoSparc | doi: 10.1038/nmeth.4169 | | v3.1 |
| Software algorithm | CTFfind4 | doi: 10.1016/j.jsb.2015.08.008 | | Version 4.1.14 |
| Software algorithm | Topaz | doi: 10.1038/s41592-019-0575-8 | | |
| Software algorithm | COOT | https://www2.mrc-lmb.cam.ac.uk/personal/pemsley/coot/ | | Version 0.9.3 |
| Software algorithm | Phenix | http://www.phenix-online.org | | Version 1.18 |
| Software algorithm | UCSF ChimeraX | https://www.cgl.ucsf.edu/chimerax/ | | Version 1.2.5 |
| Software algorithm | Molprobity | Duke Biochemistry, http://molprobity.biochem.duke.edu/ | | |
| Software algorithm | PyMol 2 | https://pymol.org/2/ | | Version 2.0.6 |
| Software algorithm | AcquireMP | Refeyn (https://www.refeyn.com) | | v.2.3 |

## Molecular cloning and plasmid generation

In order to overexpress the Aq880cHis protein, a pET-28a(+)_Aq880cHis plasmid was constructed. For this purpose, the genomic DNA of *A. aeolicus* strain was isolated and used as template for amplification of the aq_880 gene with the primer pair listed in *supplementary file 1c* as described by (1). The PCR product was inserted into the pET-28a(+) vector via *Xho*I and *Nco*I restriction sites. For all the Aq880cHis variants, site-directed mutagenesis was done to gain either the C-terminal truncated proteins or the arginine and lysine-to-alanine mutants. The polymerase chain reaction (PCR) was performed using the Platinum SuperFi PCR Master Mix (Invitrogen) and the primers listed in *supplementary file 1c*. For the pET-28a(+)_Hhal2243nHis construct, isolated chromosomal DNA of *H. halophila* was used as template, and the PCR amplified product (primers listed below) was inserted via the restriction sites *Nhe*I and *Bpu1102*I into the pET-28a(+) vector (2). For Primer and plasmid constructions, see the *Supplementary file 1c*.

## Expression and purification of recombinant proteins

### Preparation of Hhal2243

For recombinant overexpression of the HARP from *H. halophila* SL1 (Hhal2243) with N-terminal His tag, a pET28(+) Hhal2243nHis plasmid was introduced into Lemo21 (DE3) competent *E. coli* cells as well as into the Rosetta (DE3) strain. For protein expression in the Lemo21 strain, an LB broth culture supplemented with 50 µg/mL kanamycin and 34 µg/mL chloramphenicol was incubated for 6 hr at

37°C. The main culture was inoculated 1:50 with this pre-culture supplemented with 500 μM rhamnose. After 2 hr at 37°C, overexpression of the Hhal2243nHis protein was induced upon addition of IPTG (0.4 mM) and cells were incubated for another 16 hr. Protein expression in the Rosetta strain was done as described below for Aq880 and mutants thereof. Cells obtained in both expression systems were harvested by centrifugation at $2000 \times g$ for 30 min at 4°C and combined. The combined cell pellets were resuspended in NPI-20 buffer (50 mM $NaH_2PO_4$/NaOH, pH 8.0, 300 mM NaCl, and 20 mM imidazole) and disrupted by sonification (output control: 50%, duty cycle: 50%, output: 20%) in three cycles for 2 min on ice. After centrifugation (4°C, 1 hr, $10,500 \times g$), the lysate was filtered and loaded onto a 1 mL HisTrap column. Elution was performed with NPI-500 buffer (as NPI-20, but containing 500 mM imidazole) applying a linear gradient over 30 column volumes (*Figure 1— figure supplement 1A*), and HARP-containing fractions were dialyzed against thrombin cleavage buffer (10 mM Tris-HCl, pH 8.0, 100 mM KCl, 0.1 mM EDTA, 2 mM $CaCl_2$, and 10% (v/v) glycerol). The N-terminal His6-tag was then removed by digestion with thrombin ($\approx 1$ U/mg; GE Healthcare) at 4°C overnight. For removal of thrombin and further purification, the protein fractions were subjected to MonoQ anion-exchange chromatography (*Figure 1—figure supplement 1A*). The tag-free HARP was eluted with thrombin cleavage buffer containing 1 M KCl and dialyzed against 'crystallization buffer' (10 mM Tris-HCl pH 8.0, 100 mM KCl, and 0.1 mM EDTA). Protein purity was analyzed by 15% SDS-PAGE (*Figure 1—figure supplement 1B*), and successful removal of the His6-tag by thrombin digestion was verified using an $\alpha-$6His-HRP antibody (*Figure 1—figure supplement 1C*). The protein was frozen in liquid nitrogen and stored at $-80$°C. All protein concentrations were determined via Bradford assay. All protein preparations were nucleic acid-free based on absorption ratios > 1 for 280/260 nm.

## Preparation of Aq880 and mutants

Aq880 with C-terminal His6 tag (Aq880cHis) and mutants thereof were overexpressed in Rosetta (DE3) cells in LB autoinduction medium (0.2% [w/v] lactose, 0.05% [w/v] glucose) supplemented with 50 μg/mL kanamycin and 34 μg/mL chloramphenicol. Cells grown at 37°C for 18–22 hr were harvested by centrifugation at $2000 \times g$ for 30 min at 4°C. After resuspension in NPI-20 buffer (see above), cells were lysed via sonification (output control: 50%, duty cycle: 50%, output: 20%) in five cycles (each 2 min) and with cooling on ice between the cycles. After centrifugation ($10,500 \times g$ for 4°C, 1 hr), the lysate was filtered and loaded onto a 1 mL HisTrap column. Elution was performed with NPI-500 buffer applying a step gradient in 20% steps over 20 column volumes. Protein purity was analyzed using 12% stain-free TGX gels detected via the ChemiDoc MP Imaging System (BioRad). The protein was dialyzed against 'storage buffer' (10 mM Tris-HCl pH 8.0, 100 mM KCl, 10 mM $MgCl_2$, 0.1 mM EDTA, 3 mM DTT immediately added before use, and 50% [v/v] glycerol) and stored at $-20$°C. For analyzing the protein's oligomerization state, analytical SEC was performed. Beforehand, Aq880cHis was purified over a MonoQ column and eluting protein fractions were dialyzed against 'crystallization buffer' (10 mM Tris-HCl pH 8.0, 100 mM KCl, and 0.1 mM EDTA). Then 250 μL protein (0.2–2.3 mg) was loaded onto the Superose 6 10/300 GL column. To obtain a calibration curve for molecular mass estimation, protein standards (Merck Sigma-Aldrich) specified in *Figure 2— figure supplement 2* were separated on the same column.

## Cryo-EM grid preparation and data collection

To prepare cryo-EM grids, 3 μL of Hhal2243 at 100 μM concentration was applied to CF 1.2/1.3 grids (Protochips) that were glow-discharged 20 s immediately before use. The sample was incubated 30 s at 100% humidity and 10°C before blotting for 11 s with blotforce $-2$ and then plunge-frozen into a liquid ethane cooled by liquid nitrogen using a Vitrobot Mark IV (FEI). Data were acquired on a Titan Krios electron microscope (Thermo Fisher Scientific, FEI) operated at 300 kV, equipped with a K3 direct electron detector (Gatan). Movies were recorded in counting mode at a pixel size of 0.833 Å per pixel using a cumulative dose of 40 $e^-/Å^2$ and 40 frames. Data acquisition was performed using EPU two with two exposures per hole with a target defocus range of 1.5–2.4 μm.

## Cryo-EM data processing

The Hhal2243 dataset was processed in CryoSparc v3.1 (*Punjani et al., 2017*). Dose-fractionated movies were gain-normalized, aligned, and dose-weighted using Patch Motion correction. The contrast transfer function (CTF) was determined using CTFfind4 (*Rohou and Grigorieff, 2015*). A total of 52,710 particles was picked using the blob picking algorithm and used to train a model that was subsequently used to pick the entire dataset using TOPAZ (*Bepler et al., 2019*). A total of 2,749,587 candidate particles were extracted and cleaned using iterative-rounds of reference-free 2D classification. The 2,665,011 particles after 2D classification were used for ab initio model reconstruction. The particles were further iteratively classified in 3D using heterogenous refinement. The 1,736,597 particles belonging to the best-aligning particles were subsequently subjected to homogenous 3D refinement, yielding 3.37 Å global resolution and a B-factor of $-181.8$ Å$^2$.

## Model building

The reconstructed density was interpreted using COOT (*Emsley and Cowtan, 2004*); a model was built manually into the electron density of the best resolved molecule and superposed to reconstruct the symmetry mates. Model building was iteratively interrupted by real-space refinements using Phenix (*Liebschner et al., 2019*). Statistics assessing the quality of the final model were generated using Molprobity (*Chen et al., 2010*). Images of the calculated density and the built model were prepared using UCSF Chimera (*Pettersen et al., 2004*), UCSF ChimeraX (*Goddard et al., 2018*), and PyMOL.

## Mass photometry

MP experiments were performed using a OneMP mass photometer (Refeyn Ltd, Oxford, UK). Data acquisition was performed using AcquireMP (Refeyn Ltd. v2.3). MP movies were recorded at 1 kHz, with exposure times varying between 0.6 and 0.9 ms, adjusted to maximize camera counts while avoiding saturation. Microscope slides ($70 \times 26$ mm$^2$) were cleaned for 5 min in 50% (v/v) isopropanol (HPLC grade in Milli-Q H$_2$O) and pure Milli-Q H$_2$O, followed by drying with a pressurized air stream. Silicon gaskets to hold the sample drops were cleaned in the same manner and fixed to clean glass slides immediately prior to measurement. The instrument was calibrated using the NativeMark Protein Standard (Thermo Fisher Scientific) immediately prior to measurements. The concentration during measurement of Aq880, Aq880 mutants, or Hhal2243 during measurements was typically 100 nM. Each protein was measured in a new gasket well (i.e., each well was used once). To find focus, 18 μL of fresh buffer (10 mM Tris pH 8.0, 100 mM KCl, and 0.1 mM EDTA) adjusted to room temperature was pipetted into a well, the focal position was identified and locked using the autofocus function of the instrument. For each acquisition, 2 μL of diluted protein was added to the well and thoroughly mixed. For each sample, three individual measurements were performed. The data were analyzed using the DiscoverMP software.

## Pre-tRNA processing assays

Activity of recombinant HARPs was analyzed essentially as described (*Nickel et al., 2017*). Processing assays were carried out in buffer F (50 mM Tris-HCl, pH 7.0, 20 mM NaCl, and 5 mM DTT added immediately before use) supplemented with 4.5 mM divalent metal ions (usually 4.5 mM MgCl$_2$). Cleavage assays were performed with 50 or 500 nM HARP and ~5 nM 5'-$^{32}$P-labeled pre-tRNA$^{Gly}$. Enzyme and substrate were preincubated separately (enzyme: 5 min at 37°C; substrate 5 min at 55°C/5 min at 37°C). To start the reaction, 4 μL of substrate mix was added to 16 μL enzyme mix. At different time points, 4 μL aliquots were withdrawn, mixed with 2× denaturing loading buffer (0.02% [w/v] bromophenol blue, 0.02% [w/v] xylene cyanol blue, 2.6 M urea, 66% [v/v] formamide, and 2× TBE) on ice and subjected to electrophoresis on 20% denaturing polyacrylamide gels. 5'-$^{32}$P-labeled pre-tRNA$^{Gly}$ substrate and the cleaved off 5'-leader product were visualized using a Bio-Imaging Analyzer FLA3000-2R (Fujifilm) and quantified with the AIDA software (Raytest). First-order rate constants of cleavage ($k_{obs}$) were calculated with Grafit 5.0.13 (Erithacus Software) by nonlinear regression analysis. HARP working solutions, obtained by dilution from stock solutions, were prepared in EDB buffer (10 mM Tris-HCl pH 7.8, 30 mM NaCl, 0.3 mM EDTA, and 1 mM DTT added immediately before use) and kept on ice before use; ~1 μL enzyme working solution was added to the aforementioned enzyme mix ($\Sigma$16 μL). All experiments were at least performed in triplicates.

## Acknowledgements

The authors thank the cryo-EM facility at the MPI for Biophysics for generous support. The authors are grateful to Jan Schuller for critical discussions on the manuscript and cryo-EM data processing. G H thanks the Max-Planck Society for financial support.

## Additional information

### Funding

| Funder | Grant reference number | Author |
|---|---|---|
| Deutsche Forschungsge-meinschaft | HA 1672/19-1 | Roland K Hartmann |

The funders had no role in study design, data collection and interpretation, or the decision to submit the work for publication.

### Author contributions

Rebecca Feyh, Investigation, Writing - original draft; Nadine B Waeber, Simone Prinz, Pietro Ivan Giammarinaro, Investigation; Gert Bange, Funding acquisition; Georg Hochberg, Resources; Roland K Hartmann, Conceptualization, Resources, Supervision, Funding acquisition, Writing - original draft, Project administration; Florian Altegoer, Conceptualization, Validation, Investigation, Visualization, Writing - original draft, Project administration

### Author ORCIDs

Pietro Ivan Giammarinaro (iD) http://orcid.org/0000-0002-0356-8481
Florian Altegoer (iD) https://orcid.org/0000-0002-6012-9047

### Decision letter and Author response

Decision letter https://doi.org/10.7554/eLife.70160.sa1
Author response https://doi.org/10.7554/eLife.70160.sa2

## Additional files

### Supplementary files

- Supplementary file 1. Supplementary Tables.
- Transparent reporting form

### Data availability

Coordinates and structure factors have been deposited within the protein data bank (PDB) and the electron microscopy data bank (EMDB) under accession codes: 7OG5 and EMD-12878. The authors declare that all other data supporting the findings of this study are available within the article and its supplementary information files.

The following datasets were generated:

| Author(s) | Year | Dataset title | Dataset URL | Database and Identifier |
|---|---|---|---|---|
| Altegoer F, Bange G | 2021 | RNA-free Ribonuclease P from Halorhodospira halophila | https://www.rcsb.org/structure/7OG5 | RCSB Protein Data Bank, 7OG5 |
| Altegoer F, Bange G | 2021 | RNA-free Ribonuclease P from Halorhodospira halophila | https://www.emdatare-source.org/EMD-12878 | EMDataResource, EMD-12878 |

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
