## [Decision Letter]

**Acceptance summary:**

Nature has evolved remarkably different enzymes for the essential processing of 5´ ends of pre-tRNAs. The ribonucleoprotein RNase P uses its RNA component for pre-tRNA recognition and catalysis, the protein-only RNase P (PRORP) contains a pentatricopeptide repeat domain for pre-tRNA recognition and a nuclease domain for catalysis, and more recently a new family, Homolog of Aquifex RNase P (HARP), was identified. The HARPs seem mysterious as they are quite small (~23 kDa) and form oligomers. Although they appeared to possess a catalytic domain, it was unclear how they would recognize and process pre-tRNAs. Here the authors have addressed these questions by determining a cryo-EM structure of a dodecameric HARP, Hhal2243, and using the structural information to strikingly demonstrate the essential nature of the oligomerization for enzymatic activity of the Aquifex HARP, Aq880. Enzymatic activity assays with mutant enzymes identify basic residues for pre-tRNA substrate recognition, and a preliminary HARP/tRNA model suggests a possible mode for pre-tRNA recognition and catalysis by the dodecamer. The authors responded comprehensively and effectively to the prior concerns raised and included new data about the oligomerization state of the mutant enzymes. The revised manuscript is acceptable for publication.

**Decision letter after peer review:**

Thank you for submitting your article "Structure and mechanistic features of the prokaryotic minimal RNase P" for consideration by *eLife*. Your article has been reviewed by 2 peer reviewers, and the evaluation has been overseen by a Reviewing and Senior Editor.

The reviewers have discussed their reviews with one another, and we have drafted this to help you prepare a revised submission. There was strong enthusiasm for this study and almost all the requested revisions, though rather numerous, are related to improving the text and figure presentation.

Essential revisions:

1) The authors aligned their cryo-EM model with the crystal structure of the Arabidopsis PRORP1 to show that the arrangement of catalytic acidic residues is conserved in these two families, although I believe the overall structure of each protein family is distinct. To clarify this point, please explain the structural alignment more (page 9, lines 178-179). Are the folds distinct, yet the catalytic residues align? The authors should consider moving Figure S5 to the main figures. The conservation of the arrangement of catalytic residues will likely be of interest to enzymologists fascinated by the consistent geometric arrangement produced by distinct structural scaffolds.

2) The authors identified two basic residues, R125 and R129, that are located in the spike-helix region of the protein. Mutation of either or both arginine residues to alanine abrogates enzymatic activity. The authors did not include experimental data (such as mass photometry) to confirm oligomerization status of these point mutants. This is a critical control that is necessary to support the conclusion.

3) The manuscript could be considerably strengthened if elements could be provided to support the proposed model for tRNA binding. Obviously the 3D structure of complex with pre-tRNA would validate (or not) the hypothesis presented here. While this is not required for this particular publication, could authors at least mention, if they attempted to obtain such a structure and if they did so, describe what they tried. In addition, Figure 3 describing SH domain mutants should be renamed "SH domain is essential for RNase P activity" because only RNase P activity was measured and not tRNA binding. Although I agree with authors that their data very strongly suggest a requirement of the distal part of helix a6 from the SH domain for tRNA binding, their results do not directly show it. This assumption and the resulting model for HARP / tRNA interaction would be considerably strengthened if authors could directly measure pre-tRNA interactions with the one or two mutants compared to wild-type using techniques such as micro scale thermophoresis or similar techniques.

4) If the authors elect not to perform additional experiments on characterizing the pre-tRNA interaction with their RNAse P (which will not be required by *eLife*), we request additional edits. We found somewhat confusing to understand which monomers were engaging pre-tRNA, and we recommend improving the presentation of the model and how it was generated. Including the coordinates of the model as a supplement file would be important.

5) It would be helpful to establish clear and consistent nomenclature for the individual subunits and dimers and to add labels to figures. For example, Figure 1A refers to both monomer and dimer subunits and a dimer subunit is shown in Figure 1C that seems to include monomer N and N´ but a different interface is shown in Figure 2A (monomer N and N+1). Labeling the monomers and the two different types of interfaces would prevent confusion throughout the manuscript. I suggest numbering the chains in the PDB file to roughly follow the nomenclature.

6) How well do the different dimers superimpose, such as 1/1´ with 2/2´, etc. or 1/2 with 2/3, etc.? Do the interfaces or monomer-monomer arrangements suggest what causes the screw axis to form? Are there indications of why a dodecamer might be favored or a higher order oligomer prohibited?

7) The relationship of pre-tRNAs to the dodecamer in the protein-RNA model is difficult to appreciate from Figure 4 and the text. The text reads, "This framework enables the cooperation of two adjacent dimer subunits, where one dimer binds the tRNA elbow in such a manner that the pre-tRNA cleavage site, at a distance of ~45 Å, directly reaches into the active site of the neighboring dimer (Figure 4B)." Could you please clarify that the adjacent dimer subunits shown in Figures 4A and B are N/N´ and N+1/N+1´ and indicate which monomer binds the tRNA elbow and which monomer binds the 5´ end?

8) Only one recognition site and one active site are labeled in Figures 4A and B. It seems that there are two recognition sites close to each other from N and N´ monomers and therefore four recognition sites in the dimer of dimers shown in Figures 4A and B. Similarly, there should be four active sites. How did you choose which recognition sites and active sites to engage substrate? It would be helpful to provide a more detailed description of how you eliminated other possibilities. Was there a combination of recognition site and active site that more closely aligned with PRORP models? If you aligned the active sites as in Suppl. Figure 5A, would the tRNA elbow recognition sites of HARP and PRORP be near each other?

Please consider briefly discussing these points. Do you envision that all five pre-tRNA molecules are engaged via both 5´ end and T loop? Do you think the binding of one substrate molecule affects binding or cleavage of other pre-tRNAs? Do you know if it is possible to form a stable complex of the dodecamer and pre-tRNA or tRNA in solution to determine stoichiometry?

9) Part of the confusion about the model could be cured by adding labels and descriptions to Figure 4. In Figure 4A, please indicate the orientation relative to Figure 1. We expected that subunit 1 would be on the right and subunit 2 would be on the left, but it seems like PIN and PIN´ could be mislabeled. In Figure 4C, please label the features of the tRNAs: D loop, T loop, 5´ end, 3´ end. Are the model and cartoon showing the same orientation? It seems like they are not the same. Are they flipped 180°?

---

## [Author Response]

Essential revisions:1) The authors aligned their cryo-EM model with the crystal structure of the Arabidopsis PRORP1 to show that the arrangement of catalytic acidic residues is conserved in these two families, although I believe the overall structure of each protein family is distinct. To clarify this point, please explain the structural alignment more (page 9, lines 178-179). Are the folds distinct, yet the catalytic residues align? The authors should consider moving Figure S5 to the main figures. The conservation of the arrangement of catalytic residues will likely be of interest to enzymologists fascinated by the consistent geometric arrangement produced by distinct structural scaffolds.

We now describe in detail how exactly the superposition was performed. While some structural elements superpose well, some don’t but still the catalytic residues arrange in a highly similar way. It now reads: “More precisely, the β-strands 5, 6 and 8 and the α-helices 16 and 20 within the metallonuclease domain of *Arabidopsis thaliana* PRORP1 (AtPRORP1) were used for the structural superposition with β1, β4, α1 and α6 at Hhal2243, as the overall scaffold of the two metallonuclease domains is related but shows large structural deviations.” Furthermore, we have moved Figure S5 to the main text (now Figure 3) and included two panels with a full view of the two metallonuclease domains allowing an easier understanding of the structural differences. The colors of the Hhal2243 depiction have been changed to blue to generate a stronger contrast.

2) The authors identified two basic residues, R125 and R129, that are located in the spike-helix region of the protein. Mutation of either or both arginine residues to alanine abrogates enzymatic activity. The authors did not include experimental data (such as mass photometry) to confirm oligomerization status of these point mutants. This is a critical control that is necessary to support the conclusion.

We fully agree that these controls were critically lacking. We have now included mass photometry data on the R125, R129 and the 5xmut Aq880 (see Figure 4D). All variants form mainly dodecamers and show similar oligomerization behavior as the wildtype. We thus concluded in the text: “Notably, variation of positively charged residues to alanines in the SH-domain did not significantly change the oligomerization behavior of Aq880 as judged by mass photometry (Figure 4D).”

3) The manuscript could be considerably strengthened if elements could be provided to support the proposed model for tRNA binding. Obviously the 3D structure of complex with pre-tRNA would validate (or not) the hypothesis presented here. While this is not required for this particular publication, could authors at least mention, if they attempted to obtain such a structure and if they did so, describe what they tried. In addition, Figure 3 describing SH domain mutants should be renamed "SH domain is essential for RNase P activity" because only RNase P activity was measured and not tRNA binding. Although I agree with authors that their data very strongly suggest a requirement of the distal part of helix a6 from the SH domain for tRNA binding, their results do not directly show it. This assumption and the resulting model for HARP / tRNA interaction would be considerably strengthened if authors could directly measure pre-tRNA interactions with the one or two mutants compared to wild-type using techniques such as micro scale thermophoresis or similar techniques.

We have renamed the respective chapter and figure text to: “SH domain is critical for RNase P activity”.

We thank this reviewer concerning the validity of our model. We completely agree that a tRNA-bound structure would significantly strengthen our study. We have indeed tried to perform MST experiments to determine the binding affinities between tRNA and HARP. Unfortunately, our attempts were unsuccessful to date as the affinities appear rather low (µM range). In addition, we tried to use hydrogen-deuterium exchange mass spectrometry (HDX-MS) to determine the regions at Aq880 involved in tRNA binding. Unfortunately, Aq880 seems undetectable by mass spectrometry probably due to inefficient proteolytic digestion.

In addition, we tried to obtain a tRNA-bound structure of Hhal2243 by cryo-EM. Again, the low affinities of tRNA complicate structure solution. Initially, large amounts of unbound tRNA contaminated the grid preventing proper 2D classification. We are currently optimizing the conditions and could already collect a small dataset of a potential tRNA-bound state. However, assuming up to 10 binding sites with partial/differential occupancy, classification is challenging and requires collection of large datasets for extensive 2D and 3D classification.

4) If the authors elect not to perform additional experiments on characterizing the pre-tRNA interaction with their RNAse P (which will not be required by eLife), we request additional edits. We found somewhat confusing to understand which monomers were engaging pre-tRNA, and we recommend improving the presentation of the model and how it was generated. Including the coordinates of the model as a supplement file would be important.

We have thoroughly reworked the presentation of the tRNA-bound state, now presented in Figure 5 (previously Figure 4). Furthermore, we have supplied the coordinates from which the figure was generated: Aq880-tRNA.pdb. We hope we could clarify how each of the monomers likely contacts a tRNA and which part of it is potentially recognized.

5) It would be helpful to establish clear and consistent nomenclature for the individual subunits and dimers and to add labels to figures. For example, Figure 1A refers to both monomer and dimer subunits and a dimer subunit is shown in Figure 1C that seems to include monomer N and N´ but a different interface is shown in Figure 2A (monomer N and N+1). Labeling the monomers and the two different types of interfaces would prevent confusion throughout the manuscript. I suggest numbering the chains in the PDB file to roughly follow the nomenclature.

We fully agree with the reviewers on this point. We have removed “subunit” throughout the text and replaced it with “monomers” or “dimers” where appropriate. In addition, the numbering in the pdb file has been changed accordingly. Figure 1C shows monomers 2/2*, while Figure 2A shows the oligomer interface between monomer 1 and 2. We also changed the nomenclature of the second monomer and used an asterisk instead of a quotation mark to prevent confusion with the 3’- and 5’-ends of the tRNA.

6) How well do the different dimers superimpose, such as 1/1´ with 2/2´, etc. or 1/2 with 2/3, etc.? Do the interfaces or monomer-monomer arrangements suggest what causes the screw axis to form? Are there indications of why a dodecamer might be favored or a higher order oligomer prohibited?

The different dimers superpose very well with r.m.s.d.’s of 0.2 or lower. We have now put more effort into making plausible why oligomerization terminates after a single screw turn when dimer 6/6* collides with dimer 1/1* (see Results section, 2nd paragraph). In addition, we have added a supplementary figure illustrating the steric clash between dimers 1/1* and 6/6* along with geometric scenarios that would allow a continuation of oligomerization (Figure 1—figure supplement 4).

7) The relationship of pre-tRNAs to the dodecamer in the protein-RNA model is difficult to appreciate from Figure 4 and the text. The text reads, "This framework enables the cooperation of two adjacent dimer subunits, where one dimer binds the tRNA elbow in such a manner that the pre-tRNA cleavage site, at a distance of ~45 Å, directly reaches into the active site of the neighboring dimer (Figure 4B)." Could you please clarify that the adjacent dimer subunits shown in Figures 4A and B are N/N´ and N+1/N+1´ and indicate which monomer binds the tRNA elbow and which monomer binds the 5´ end?

We have changed the figure (now Figure 5) and the text accordingly: “This framework enables the cooperation of two adjacent dimers, where one monomer of each dimer binds the tRNA elbow in such a manner that the pre-tRNA cleavage site, at a distance of ~45 Å, directly reaches into the active site of the monomer from the neighboring dimer. According to this scenario and as exemplarily illustrated in Figure 5B, the SH domain of monomer 6* would bind the tRNA elbow region, while the tRNA 5’-end is docked into the active site of monomer 5*. This way, five pre-tRNAs can potentially be docked onto the upper or lower layer of the dodecameric, distorted double donut structure (Figure 5C)."

8) Only one recognition site and one active site are labeled in Figures 4A and B. It seems that there are two recognition sites close to each other from N and N´ monomers and therefore four recognition sites in the dimer of dimers shown in Figures 4A and B. Similarly, there should be four active sites. How did you choose which recognition sites and active sites to engage substrate? It would be helpful to provide a more detailed description of how you eliminated other possibilities. Was there a combination of recognition site and active site that more closely aligned with PRORP models? If you aligned the active sites as in Suppl. Figure 5A, would the tRNA elbow recognition sites of HARP and PRORP be near each other?

We have only illustrated one recognition site and one active site in Figure 5A and B (formerly Figure 4) to keep the figure as simple as possible. As stated above, the elbow region of tRNA 5* is recognized by monomer 6* while the cleavage site reaches into the active site of monomer 5*. Monomers 5 and 6 would recognize tRNA 5 in a mirrored fashion (monomer 5: elbow and monomer 6: active site). In total, 10 active sites out of 12 could be potentially occupied simultaneously. The asymmetric screw-like assembly sterically excludes that dimers 1 and 6 together are able to productively bind a tRNA.

Furthermore, we have superposed HARP and AtPRORP but in the latter, metallonuclease domain and PPR-domain are connected via a flexible linker, which complicates a direct comparison.

Please consider briefly discussing these points. Do you envision that all five pre-tRNA molecules are engaged via both 5´ end and T loop? Do you think the binding of one substrate molecule affects binding or cleavage of other pre-tRNAs? Do you know if it is possible to form a stable complex of the dodecamer and pre-tRNA or tRNA in solution to determine stoichiometry?

We can largely rule out a cooperative tRNA binding mode based on comparison of single and multiple turnover kinetics using Aq880 (Figure 2D, E in Nickel et al., PNAS 2017) We, however, consider it likely that the dimers are more rigid in the context of a higher oligomer structure, thereby supporting productive pre-tRNA binding.

We haven't been able to form a stable HARP:tRNA complex on size exclusion columns. The low affinity in the µ-molar range prevents an analysis by mass photometry, as we use 50- 100 nanomolar concentration in this setup. Concerning the maximum amount of bound tRNAs, we consider it possible that a “fully loaded” state may exist in vivo, considering that the tRNA concentration has been estimated to be 0.5 mM in an *E. coli* cell.

We have cautiously discussed these points in the revised version of the manuscript.

9) Part of the confusion about the model could be cured by adding labels and descriptions to Figure 4. In Figure 4A, please indicate the orientation relative to Figure 1. We expected that subunit 1 would be on the right and subunit 2 would be on the left, but it seems like PIN and PIN´ could be mislabeled. In Figure 4C, please label the features of the tRNAs: D loop, T loop, 5´ end, 3´ end. Are the model and cartoon showing the same orientation? It seems like they are not the same. Are they flipped 180°?

We have now thoroughly reworked Figure 5 (previously Figure 4). Figure 5C (left) is now in the same orientation as Figure 5A. The orientation of 5C relative to Figure 1 has been added as sketch in the upper left corner. Furthermore, the nomenclature has been changed to be consistent. The tRNA has been labeled accordingly. We hope that our changes make it easier to follow our conclusions.